# Identification of the fungal ligand triggering cytotoxic PRR-mediated NK cell killing of *Cryptococcus* and *Candida*

Shu Shun Li[1,2], Henry Ogbomo[1,2], Michael K. Mansour[3], Richard F. Xiang[1,2], Lian Szabo[4], Fay Munro[5], Priyanka Mukherjee[5], Roy A. Mariuzza[6], Matthias Amrein[5], Jatin M. Vyas [3], Stephen M. Robbins[7,8] & Christopher H. Mody [1,2,4]

Natural killer (NK) cells use the activating receptor NKp30 as a microbial pattern-recognition receptor to recognize, activate cytolytic pathways, and directly kill the fungi *Cryptococcus neoformans* and *Candida albicans*. However, the fungal pathogen-associated molecular pattern (PAMP) that triggers NKp30-mediated killing remains to be identified. Here we show that β-1,3-glucan, a component of the fungal cell wall, binds to NKp30. We further demonstrate that β-1,3-glucan stimulates granule convergence and polarization, as shown by live cell imaging. Through Src Family Kinase signaling, β-1,3-glucan increases expression and clustering of NKp30 at the microbial and NK cell synapse to induce perforin release for fungal cytotoxicity. Rather than blocking the interaction between fungi and NK cells, soluble β-1,3-glucan enhances fungal killing and restores defective cryptococcal killing by NK cells from HIV-positive individuals, implicating β-1,3-glucan to be both an activating ligand and a soluble PAMP that shapes NK cell host immunity.

[1] Department of Microbiology, Immunology and Infectious Diseases, University of Calgary, Calgary T2N 4N1, Canada. [2] The Calvin, Phoebe and Joan Snyder Institute for Chronic Diseases, University of Calgary, Calgary T2N 4N1, Canada. [3] Department of Medicine Division of Infectious Diseases, Massachusetts General Hospital, Boston, MA 02114, USA. [4] Department of Medicine, University of Calgary, Calgary T2N 4N1, Canada. [5] Department of Cell Biology and Anatomy, University of Calgary, Calgary T2N 4N1, Canada. [6] Department of Cell Biology & Molecular Genetics, University of Maryland, College Park, MD 20742, USA. [7] Department of Biochemistry and Molecular Biology, University of Calgary, Calgary T2N 4N1, Canada. [8] Southern Alberta Cancer Research Institute, University of Calgary, Calgary T2N 4N1, Canada. Shu Shun Li and Henry Ogbomo contributed equally to this work  Correspondence and requests for materials should be addressed to C.H.M. (email: cmody@ucalgary.ca)

Invasive fungal infections are both widespread and increasing in frequency, particularly in immunocompromised individuals[1,2]. *Cryptococcus neoformans* (*C. neoformans*) causes life-threatening meningitis and pneumonia in AIDS and other immunosuppressed patients, while *Candida albicans* (*C. albicans*) produces a devastating mycosis in post-surgical and critically ill patients[3,4]. Understanding host defense for developing more effective treatment or control of fungal invasive infections remains a major challenge.

Innate immune host defense is evolutionarily conserved, and has an essential function in immunity against microbial infections. Natural killer (NK) cells are innate cytotoxic cells that directly recognize and kill *C. neoformans* and *C. albicans*. In mice, NK cells are required for optimal clearance of *C. neoformans* and *C. albicans*[5–9]. In humans, NK cells bind to *C. neoformans* and mediate killing[10,11]. By contrast, NK cell function is defective in patients with *cryptococcal meningitis*[12], and NK cells from HIV-infected patients are unable to kill *C. neoformans*[13]. NK cells depleted of the critical effector molecule, perforin, could be seen in contact with *Cryptococcus* within the brain of a patient that succumbed to the infection[14]. Previously, we demonstrated that the NK cell receptor, NKp30, is the pattern-recognition receptor (PRR) recognizing *C. neoformans* and *C. albicans* that triggers activation of PI3K and Erk 1/2, perforin release, and fungal cytotoxicity[15].

PRRs are proteins expressed by cells of the immune system that recognize pathogen-associated molecular patterns (PAMPs) as danger signals. PRR were previously organized into two categories. Phagocytic PRRs, such as Dectin-1, MARCO, scavenger receptor A, and mannose receptors, are expressed by macrophages, dendritic cells, monocytes, and neutrophils, and activate phagocytosis upon binding of a microbial PAMP[16–19]. Signaling PRR are transmembrane or cytoplasmic receptors that stimulate gene transcription of pro-inflammatory cytokines, type I interferons, chemokines, antimicrobial peptides, and costimulatory molecules in a wide variety of immune and non-immune cells. Signaling PPRs include extracellular Toll-like receptors, C-type lectin receptors, intracellular nucleotide-binding oligomerization domain-like receptors (NLR), and retinoic acid inducible gene I-like helicase receptors (RLR)[20]. In addition to these categories, a new class of PRR has been described that includes NK cell-activating receptors, NKp30, NKp46, and CD56 that bind to fungi and parasites to induce mobilization and release of cytotoxic granules that kill the pathogen[15,21–23]. NKp30, NKp46, and CD56 are all members of the immunoglobulin-like transmembrane receptor family that use ITAM-containing adaptor proteins to signal. Studies demonstrating direct binding to fungal and parasitic PAMPs suggest that Ig-like family members that activate NK cells for microbial killing be added to the PRR families forming a cytotoxic PRR subfamily. Although a PAMP for NKp46 has been identified[21], the microbial PAMP for the cytotoxic PRR NKp30 remains to be identified.

PAMPs often serve as an essential function in the pathogen and are often shared among entire classes of microbes. Molecules expressing PAMPs are either structural determinants or required for virulence[24]. The structure of *C. neoformans* consists of a unique polysaccharide capsule that surrounds the organism[25]. Beneath the capsule is the cell wall and membrane. The cell wall consists of a complex organization of polysaccharides, with smaller amounts of proteins, lipids, and pigments, that are directly exposed in *C. albicans* and acapsular *C. neoformans*, and to a lesser extent by encapsulated *Cryptococcus*[26,27]. Polysaccharides constitute nearly 80% of the fungal dry weight and include glucuronoxylomannan and galactoxylomannan in the capsule, and glucans, chitin, and chitosan in the cell wall. NKp30 mediates NK cell killing of both acapsular *Cryptococcus* and *C.*

*albicans* as well as encapsulated *C. neoformans*; therefore, the potential ligand for NKp30 is not a component of the cryptococcal capsule. Additionally, *Cryptococcus* (phyla Basidiomycota) is separated from *C. albicans* (phyla Ascomycota) by 400 million years of evolution[28], suggesting that the ligand for NKp30 is essential and preserved among widely divergent phyla. Since glucans are major structural components of fungal cell walls, our focus was narrowed to a limited subset of β-glucans that were the most likely candidates for the NKp30 ligand.

We used a variety of approaches including antibody detection and atomic force spectroscopy to demonstrate that soluble and immobilized β-1,3-glucan binds NKp30. We found that β-1,3-glucan induces Src family kinase signal transduction, synapse formation, and cytotoxic granule trafficking as seen by live cell imaging. β-1,3-glucan is necessary for killing, using fungi treated with an echinocandin as a loss-of-function approach. Surprisingly, soluble β-1,3-glucan enhances receptor and effector molecule expression and enhances killing in NK cells from healthy as well as HIV-infected patients with defective antifungal activity.

## Results

**β-1,3-glucan binds to NK cells.** Since the same receptor, NKp30, mediates NK cell recognition and killing of *C. neoformans* and *C. albicans*, it was likely that the fungal ligand is shared between these widely divergent phyla. We reasoned that an essential structural molecule was the likely candidate. As such, α-glucans and β-glucans are the major structural components of fungal cell walls. Since the former is not found in the cell wall of *C. albicans*[29], α-glucan was excluded from our list of candidates as a ligand for NKp30. The latter includes β-1,3-glucan, β-1,4-glucan, β-1,6-glucan, or mixes of β-1,3-/β-1,4- or β-1,6-glucan. However, *C. albicans* and *C. neoformans* share only β-1,3-glucan and β-1,6-glucan[29,30], which narrowed our focus.

Experiments were performed to examine whether β-glucans could bind to YT cells, an NK cell line that kills *Cryptococcus* and *Candida*[15]. YT cells were incubated with a preparation of the cryptococcal cell wall/membrane (CCW/M)[31]. Antibody labeling and flow cytometry demonstrated β-1,3-glucan on the surface of YT cells (Fig. 1a). To test if soluble β-glucans bound to YT cells, the cells were incubated with laminarin (derived from *Laminaria digitata*), consisting of β-1,3-glucan with β-1,6 branches (ratio ~3:1)[32]. β-1,3-glucan was detected on YT cells (Fig. 1b, left panel), but not on untreated YT cells (Fig. 1b, right panel). Since β-glucan from *L. digitata* and *Saccharomyces cerevisiae* (*S. cerevisiae*) is a mixture of β-1,3 and β-1,6-glucans, experiments were performed to determine whether the binding was due to β-1,3–glucan. For this purpose, the experiment was repeated with soluble laminarihexaose produced from curdlan, which comprises only β-1,3-glucan or pustulan, comprising only β-1,6-glucan. We found that laminarihexaose bound to YT cells (Fig. 1c), while pustulan did not (Fig. 1d), indicating that β-1,3-glucan is sufficient for binding to YT cells.

**β-1,3-glucan binds to recombinant NKp30.** To determine whether the immobilized β-glucan bound to NKp30, polystyrene beads were conjugated with or without β-glucan derived from *S. cerevisiae* (Supplementary Fig. 1A-B)[33], which consists of β-1,3-glucan with β-1,6 branches (ratio ~5:1)[29]. Binding of a recombinant Fc-NKp30 fusion protein to β-glucan-conjugated beads (β-1,3-GB) was revealed by flow cytometry using a polyclonal anti-NKp30 antibody (Fig. 1e).

1C01 is a monoclonal antibody (mAb) that binds to NKp30 and blocks the recognition of fungi[15]. We questioned whether this antibody would block the interaction of NKp30 with its potential binding partner, β-1,3-glucan. To test this, NKp30-Fc was

incubated with or without 1C01, and the mixture was incubated with β-1,3-GB. We were unable to detect the presence of NKp30 on β-1,3-GB using a polyclonal anti-NKp30 antibody, suggesting that mAb 1C01 blocked the binding of NKp30-Fc to β-1,3-GB (Fig. 1f). We considered the possibility that 1C01 might block the binding site for polyclonal anti-NKp30; however, this was not the case as YT cells demonstrated similar labeling with polyclonal anti-NKp30 with or without pre-incubation with 1C01 (Supplementary Fig. 1C). This reveals that 1C01 was bound to, or resulted in, steric hindrance of the β-1,3-glucan binding site on NKp30.

Both *C. neoformans* and *C. albicans* express β-1,3-glucan in their cell wall. To determine whether the ectodomain of NKp30

could bind to these two microorganisms, we incubated recombinant NKp30 ectodomain with *C. neoformans* or *C. albicans*, and labeled them with anti-NKp30 Ab. NKp30 was detected by flow cytometric analysis (Fig. 1g) consistent with NKp30 binding to β-1,3-glucan in these two fungi.

To determine whether β-1,3-glucan could immunoprecipitate NKp30 from YT cells, protein G beads were conjugated with an mAb against β-1,3-glucan. The beads were incubated with a YT cell lysate that had been incubated with β-1,3-glucan (laminarin). Western blotting showed that NKp30 was immunoprecipitated by β-1,3-glucan bound to anti-β-1,3-glucan mAb-coated beads (Fig. 1h, 2nd lane from left), providing additional evidence that NKp30 is the binding partner for β-1,3-glucan.

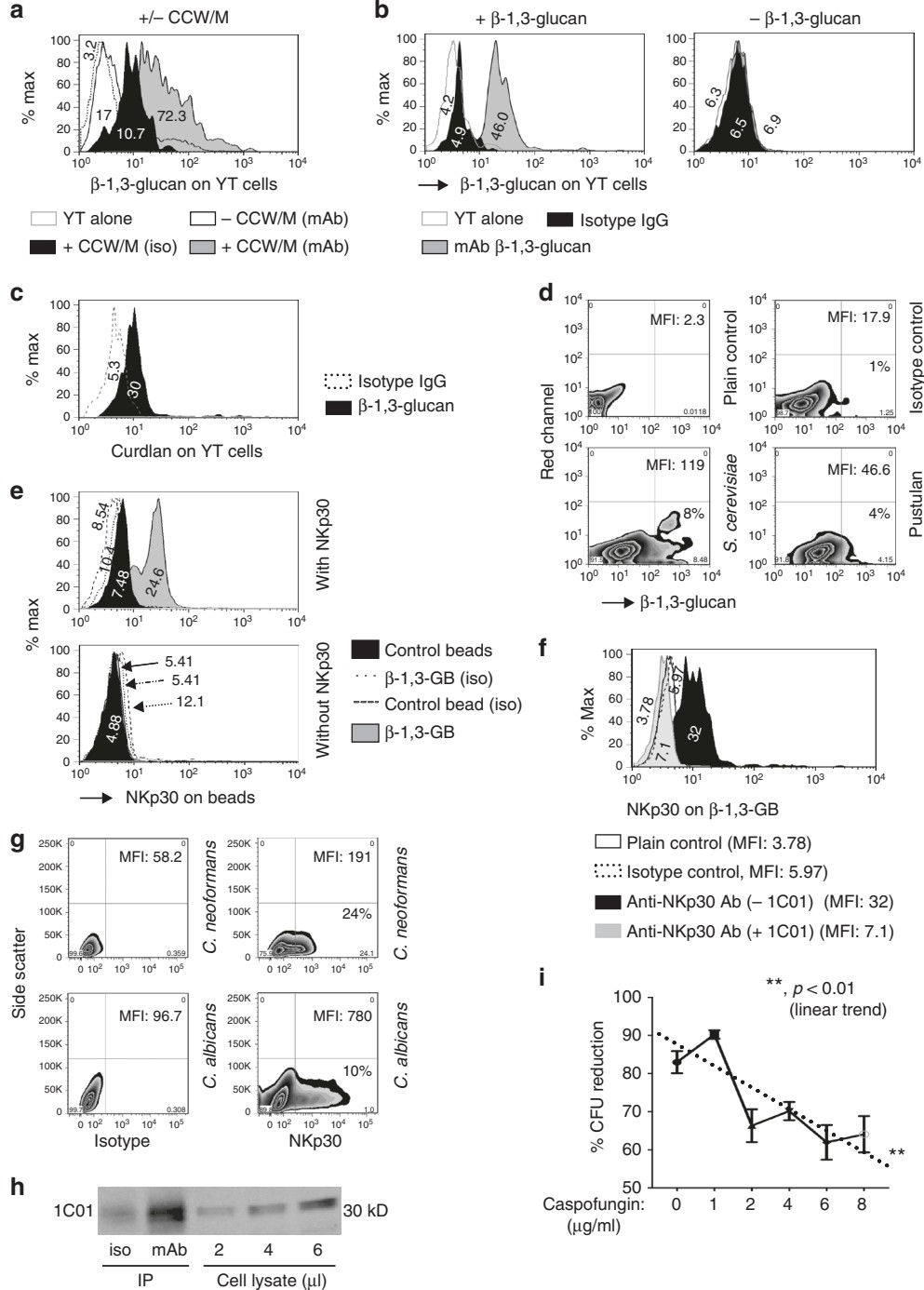

**β-1,3-glucan is required for NK killing of *Cryptococcus*.** To explore whether β-1,3-glucan was required for NK cell killing of *Cryptococcus*, we took advantage of the ability of caspofungin to interrupt β-1,3-glucan synthesis, despite the relative resistance of *C. neoformans* to this agent[34,35]. *C. neoformans* (B3501) was treated with various concentrations of caspofungin and NK cell killing was examined. As expected, caspofungin reduced the percentage of β-1,3-glucan-positive *C. neoformans* (strain B3501, Supplementary Fig. 2a) and expression of β-1,3-glucan (Supplementary Fig. 2B). While there was a modest effect on the growth of *C. neoformans* (Supplementary Fig. 2C), YT cell anticryptococcal activity was reduced after treatment with caspofungin in a dose-dependent manner (Fig. 1i and Supplementary Fig. 2C, triangles) revealing that β-1,3-glucan was required for NK cell host defense against *Cryptococcus*.

**β-1,3-glucan enhances NK cell killing in HIV patients.** Since β-1,3-glucan bound to NKp30 and was required for NK cell killing, we speculated that free β-1,3-glucan would bind and block NKp30 receptors on NK cells and abrogate NK killing. To test this hypothesis, we pretreated YT cells with soluble laminarin before the addition of *C. neoformans*. To our surprise, β-1,3-glucan did not block, but instead, elicited dose-dependent enhanced YT cell killing of *C. neoformans* (Fig. 2a). Having previously shown that NKp30 mediates NK cell killing of *C. albicans*[15], the experiments were performed to determine whether β-1,3-glucan would also enhance the killing of *C. albicans*. We found that β-1,3-glucan stimulated a dose-dependent increase in YT cell killing of *C. albicans* (Fig. 2b). β-1,3-glucan from other sources similarly enhanced YT cell killing of *C. neoformans* (Supplementary Fig. 3A-B). Notably, β-1,3-glucan showed no impact on YT cell viability (> 95%) or growth of *Cryptococcus* alone (Supplementary Fig. 3C). The β-1,3-glucans used in this study were derived from several sources including *L. digitata* and *S. cerevisiae* that contain glucans with β-1,3- and β-1,6- branches. To examine whether β-1,6-glucan contributed to the enhanced killing, YT cells were incubated with pustulan, which contains only β-1,6-glucans. Treatment of YT cells with β-1,6-glucans failed to enhance killing (Supplementary Fig. 3D).

We also tested the ability of β-1,3-glucan to enhance cytotoxicity of primary NK cells against *Cryptococcus*. Primary NK cells from healthy individuals were pretreated with β-1,3-glucan as described above, and similarly, killing by primary NK cells was enhanced (Fig. 2c). Since NK cells from HIV-infected patients have defective killing of *C. neoformans* and this defect can be restored by using interleukin-12[15,36], we wondered if β-

1,3-glucan might have a similar effect. β-1,3-glucan enhanced killing of *Cryptococcus* by NK cells from HIV-infected individuals (Fig. 2d).

**β-1,3-glucan binds NK cells as shown by live cell AFM.** To explore the mechanism by which β-1,3-glucan enhanced and restored defective NK cell cytotoxicity, we characterized NK cell binding to β-1,3-glucan using an atomic force microscope to perform single-cell force spectroscopy (SCFS). This method has the advantage of being able to measure the binding force of cell surface receptors with their ligands on live cells during controlled physiologic conditions (temperature, pH, etc.)[37]. A single YT cell was fixed to a cantilever tip, and purified β-1,3-glucan, mannan (α-1,2, α-1,3-, and α-1,6-D-mannose in fungi), or pustulan was conjugated to the glass surface (coating in Supplementary Fig. 4A and schematic drawing for binding in Supplementary Fig. 4B). Force–distance (F–D) curves from a single YT cell detaching from β-1,3-glucan required 2000–3000 pN (Fig. 3a, third panel from left). Heat-killed *S. cerevisiae*, which was the source of β-1,3-glucan used in some experiments, were adhered to glass to make a nearly confluent single cell layer as illustrated (Supplementary Fig. 4C, F). A representative F–D curve showed the force used to detach a single YT cell from *S. cerevisiae* was greater than from control (470 ± 141 vs. 45 ± 11 pN), but less than that from purified β-1,3-glucan (Fig. 3a, the fourth panel from left), suggesting that β-1,3-glucan could account for the binding force between YT cells and *Saccharomyces* cells. The accumulated retraction forces from multiple touches (Fig. 3b, left panel) and mean retraction forces (Fig. 3b, right panel) to the β-1,3-glucan-conjugated surface were significantly higher than to the controls ($p < 0.01$, by one-way ANOVA). The binding forces of YT cells to β-1,3-glucan were comparable to previously reported receptor ligands such as LFA-1 and ICAM-1[38,39], and to H7-B6 (Fig. 3c), the canonical ligand for NKp30 expressed by K562 tumor cells, while there was only minimal binding to mannan or pustulan (Fig. 3a–c).

In contrast to *S. cerevisiae*, the distribution of glucans in the cell wall of *Cryptococcus* is reversed so that β-1,6-glucan exceeds β-1,3-glucan. While we observed strong binding of YT cells to β-1,3-glucan and to *S. cerevisiae*, we asked whether β-1,3-glucan of *C. neoformans* was required for NK cell binding. *C. neoformans* (strain B3501) were pretreated with caspofungin to reduce β-1,3-glucan synthesis (Supplementary Fig. 2A-B). A single wild-type fungal cell (B3501) or β-1,3-glucan-depleted B3501 (B3501$^{\Delta glucan}$) cell was glued to a cantilever and YT cells were adhered to the glass surface (schematic in Supplementary Fig. 4D-E, F). This

**Fig. 1** β-1,3-glucan is required for NKp30-mediated killing of *Cryptococcus*. **a** Flow cytometric analysis of cryptococcal cell wall/membrane (CCW/M) binding to YT cells as detected by mAb to β-1,3-glucan vs. isotype control Ab. **b** Flow cytometric analysis of β-glucan binding to YT cell. YT cells were incubated with (left panel) or without (right panel) laminarin, a linear β-1,3-glucan with β-1,6-linkages. **c** Flow cytometric analysis of laminarihexaose binding to YT cells. YT cells were incubated with laminarihexaose (from curdlan, comprising only β-1,3-glucan). **d** Comparison of binding of pustulan vs. β-1,3-glucan to YT cells analyzed using flow cytometry. YT cells were incubated with pustulan (comprising only β-1,6-glucan) or β-1,3-glucan (derived from *S. cerevisiae*). This experiment was performed twice. **e** Flow cytometric analysis of a recombinant NKp30-Fc fusion protein binding to β-glucan-conjugated beads (β-1,3-GB) compared to unconjugated polystyrene beads as control. NKp30-Fc on the beads was detected with anti-NKp30 antibody (1C01). **f** Anti-NKp30 mAb (1C01) blocked NKp30 binding to β-1,3-GB. Recombinant NKp30 was incubated with 1C01 before being applied to beads conjugated with β-1,3-glucan. The presence of NKp30 on β-1,3-GB was detected using the polyclonal anti-NKp30 antibody. **g** NKp30 binding to *C. neoformans* vs. *C. albicans* analyzed using flow cytometry. The experiment was performed twice. **h** Immunoprecipitation of NKp30 with β-1,3-glucan. YT cell lysate was incubated with β-1,3-glucan (laminarin) before being incubated with protein G beads that had been conjugated with a mAb against β-1,3-glucan. **i** YT cell killing of *C. neoformans* (B3501) treated with caspofungin. Caspofungin concentrations were as indicated. % reduction in CFU = CFU (B3501 with caspofungin alone) − CFU (B3501 with corresponding caspofungin plus YT cells)/CFU (B3501 with caspofungin alone) × 100 from raw data (Supplementary Fig. 3C). **, $p < 0.01$. Data were analyzed using one-way ANOVA. All experiments were repeated three to five times on different days with similar results, unless specified otherwise. For flow cytometry analysis of 1,3-glucan, a mAb against β-1,3-glucan (Biosupplies, #400-2, Australia) and FITC-labeled secondary Ab were used. β-1,3-GB polystyrene beads conjugated with β-1,3-glucan, unconjugated beads polystyrene beads without β-1,3-glucan conjugation, rNKp30 recombinant NKp30 protein, anti-glucan mAb anti-β-1,3-glucan, IP immunoprecipitate, cell lysate raw cell lysate, iso IgG isotype control

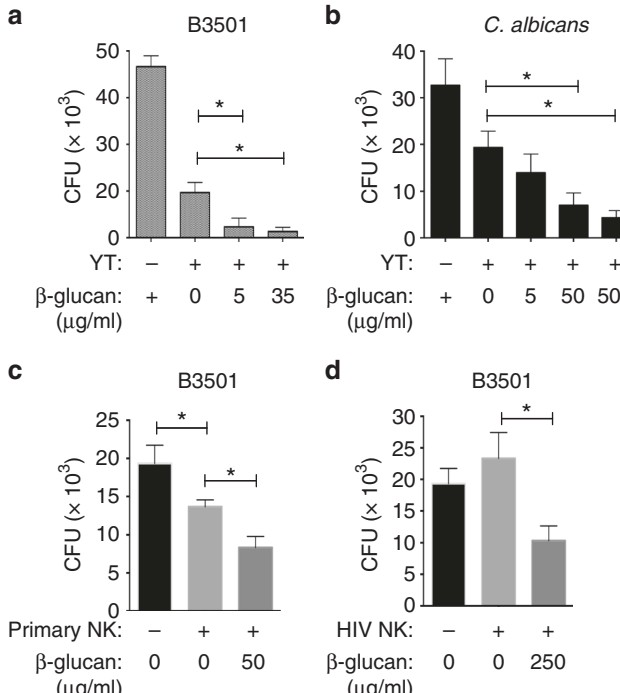

**Fig. 2** β-1,3-glucan enhances killing by NK cells from healthy subjects and HIV patients. **a** YT cell killing of *C. neoformans* in the presence of β-1,3-glucan (CFU). YT cells were treated with 0, 5, and 35 μg/ml of soluble β-1,3-glucan (laminarin derived from *L. digitata*) for 4 h before the addition of *C. neoformans* (strain B3501). CFU were determined in quadruplicate ± SEM. **b** YT cell killing of *C. albicans* in the presence of increasing doses of β-1,3-glucan (0, 5, 50, and 500 μg/ml). **c** Primary NK cell killing of *C. neoformans* (B3501) in the presence of β-1,3-glucan (0 and 50 μg/ml) for NK cells from healthy subjects. **d** Killing of *C. neoformans* (B3501) by primary NK cells from HIV-infected individuals in the presence of β-1,3-glucan (0 and 250 μg/ml). All experiments were performed for three to five times with similar results. *, *p* < 0.05. Data were analyzed using one-way ANOVA

resulted in a marked reduction in the binding force to B3501^Δglucan (Fig. 3d, middle and right panels) compared to wild-type B3501 (Fig. 3d, left. *p* < 0.01, by *T*-test), revealing that β-1,3-glucan is required for binding of *C. neoformans* to YT cells. To further confirm that β-1,3-glucan binds to YT cells via NKp30, siRNA was used to knock down NKp30 in YT cells[15], and binding of β-1,3-glucan-coated beads (β-1,3-GB) to these YT cells was measured by SCFS/atomic force microscopy(AFM). Accumulated binding forces showed that NKp30 knockdown abrogated binding of β-1,3-GB to YT cells (Fig. 3e). It is known that other receptors that may recognize β-1,3-glucan, although only dectin-1 has been identified as the primary receptor on leukocytes (reviewed in ref.[40]). Our data showed that dectin-1 is not expressed by YT cells or primary NK cells (Supplementary Fig. 5A). Taken together, we conclude that the binding of β-1,3-glucan to YT cells is through NKp30.

**β-1,3-glucan binds to NKp30 as demonstrated by AFM**. Having demonstrated that β-1,3-glucan binds to YT cells and that β-1,3-glucan is required for binding of *C. neoformans* to NK cells, we sought to determine whether NKp30 was sufficient for binding using AFM. To do so, β-1,3-GB were fixed to a cantilever tip and a glass surface was coated with the recombinant ectodomain of NKp30 (schematic in Supplementary Fig. 4G). Representative F–D curves (Fig. 3f), accumulated binding forces (Fig. 3g, left

panel), and mean forces (Fig. 3g, right panel) showed that NKp30 bound with much greater force to β-1,3-GB than to controls (*p* < 0.01, by one-way ANOVA). These data reveal that NKp30 was sufficient to explain the binding force, and is a major binding partner for β-1,3-glucan.

In some experiments, a recombinant NKp30-Fc fusion protein was used. We considered the possibility that β-1,3-glucan bound to the Fc portion of this fusion protein. To test for this possibility, we compared binding to another recombinant Fc fusion protein, CD28-Fc. AFM showed that binding forces of CD28-Fc were comparable to other controls, while forces for NKp30-Fc was significantly higher (Fig. 3h, *p* < 0.01, by *T*-test) indicating β-1,3-glucan binding to NKp30-Fc is through NKp30 rather than Fc.

**β-1,3-glucan induces granule polarization in NK cells**. NK cell killing of *Cryptococcus* is a perforin-mediated process[15], in which granules traffic to the immunological synapse (IS) to release their cytolytic cargo and kill the target cells. To determine whether β-1,3-glucan was sufficient to elicit NK cell granule trafficking in real-time, YT cells transfected with microtubule-associated protein 4-GFP (YT-MAP4-GFP cells) were labeled with Lysotracker Red DND-99 to visualize granule trafficking, and β-1,3-GB were used as the stimulus. Time-lapse video of YT cells in contact with a bead was obtained to capture granule movement. Granules in the YT cell converged toward one another (Fig. 4a, upper two rows, Fig. 4b and Supplementary movie 1) and polarized toward the β-1,3-GB (Fig. 4a, upper two rows, Fig. 4c and Supplementary movie 1), but not to control beads (Fig. 4a, third row, and Fig. 4b, c and Supplementary movies 2-3). To confirm that NKp30 was required for YT cell granule polarization toward β-1,3-glucan, siRNA was used to knock down NKp30 in YT cells, and live cell imaging was performed. YT cells with reduced expression of NKp30 failed to converge and polarize granules in response to β-1,3-GB (Supplementary movie 7) compared with wild-type YT cells, indicating that NKp30 is critical for the YT cell response to β-1,3-glucan.

**β-1,3-glucan stimulates perforin polarization and release**. When NK cells form a synapse with *Cryptococcus*, perforin is polarized to the IS[15]. To determine whether β-1,3-glucan was sufficient to stimulate perforin polarization, YT cells were incubated with β-1,3-GB, and labeled with anti-perforin antibody. Representative photomicrographs (Fig. 4d) and measurements of the distance between perforin and the IS (Fig. 4e) showed that perforin was polarized at the IS in more than 80% of the YT cells in contact with β-1,3-GB, but not to control beads (Fig. 4f). These data support the conclusion that NK cells polarize perforin to the IS in response to β-1,3-glucan as it does during fungal killing.

We previously showed that perforin expression is markedly increased after stimulation with *C. neoformans*[15,41]. To determine whether β-1,3-glucan was sufficient to increase perforin expression, YT cells were incubated with soluble β-1,3-glucan. We found that both soluble β-1,3-glucan and immobilized β-1,3-GB enhanced perforin levels (Fig. 4g). Soluble and immobilized β-1,3-glucan also resulted in increased perforin levels in the culture media (Fig. 4h), suggesting increased degranulation and release.

**β-1,3-glucan stimulates NKp30 clustering at the IS**. In addition to increased perforin expression, there are a number of possible mechanisms by which β-1,3-glucan could enhance NK cell effector function including increased receptor expression, synapse formation, and the magnitude of signaling events. We examined NKp30 expression after stimulation by β-1,3-glucan. Flow cytometry showed that β-1,3-glucan produced a time-dependent increase in NKp30 expression on YT cells (Fig. 5a, b, and

Supplementary Fig. 5B-C). This provides a possible mechanism, whereby greater signaling occurs because of enhanced receptor expression.

We previously showed that *Cryptococcus* forms a synapse with NK cells, which includes clustering of NKp30 at the area of contact prior to perforin polarization[15]. To determine whether NKp30 clustered in response to β-1,3-glucan, YT cells were mixed with β-1,3-GB and labeled with anti-NKp30 Ab. Photomicrographs showed that NKp30 was clustered at the synapse between YT and β-1,3-GB (Fig. 5c, d). More than 80% of the YT cells in contact with β-1,3-GB had NKp30 clustering (Fig. 5e), revealing that β-1,3-glucan not only induced NKp30 expression, but also induced its relocation to the IS, which would provide a signaling platform that enhanced NK cell killing.

To determine the proximity of clustered NKp30 to the IS, we used total internal reflection fluorescence (TIRF) microscopy to visualize events at a pre-defined distance (80–100 nm) from the interface between NK cells and β-1,3-glucan. In conjunction with TIRF, spinning disc (SD) confocal microscopy was used to capture images from the top to the bottom of the cells. YT cells were loaded into a glass chamber that had been coated with β-1,3-glucan or mannan (Supplementary Fig. 4A). YT cells that came into contact with β-1,3-glucan or control surfaces showed 1C01 labeling throughout the cells using SD; however, only the cells in contact with the β-1,3-glucan-coated surface showed increased 1C01 labeling at the interface by TIRF microscopy (Fig. 5f). Quantitative image analysis of the pixel intensities from NKp30 labeling was performed as illustrated (Supplementary Fig. 6A-B).

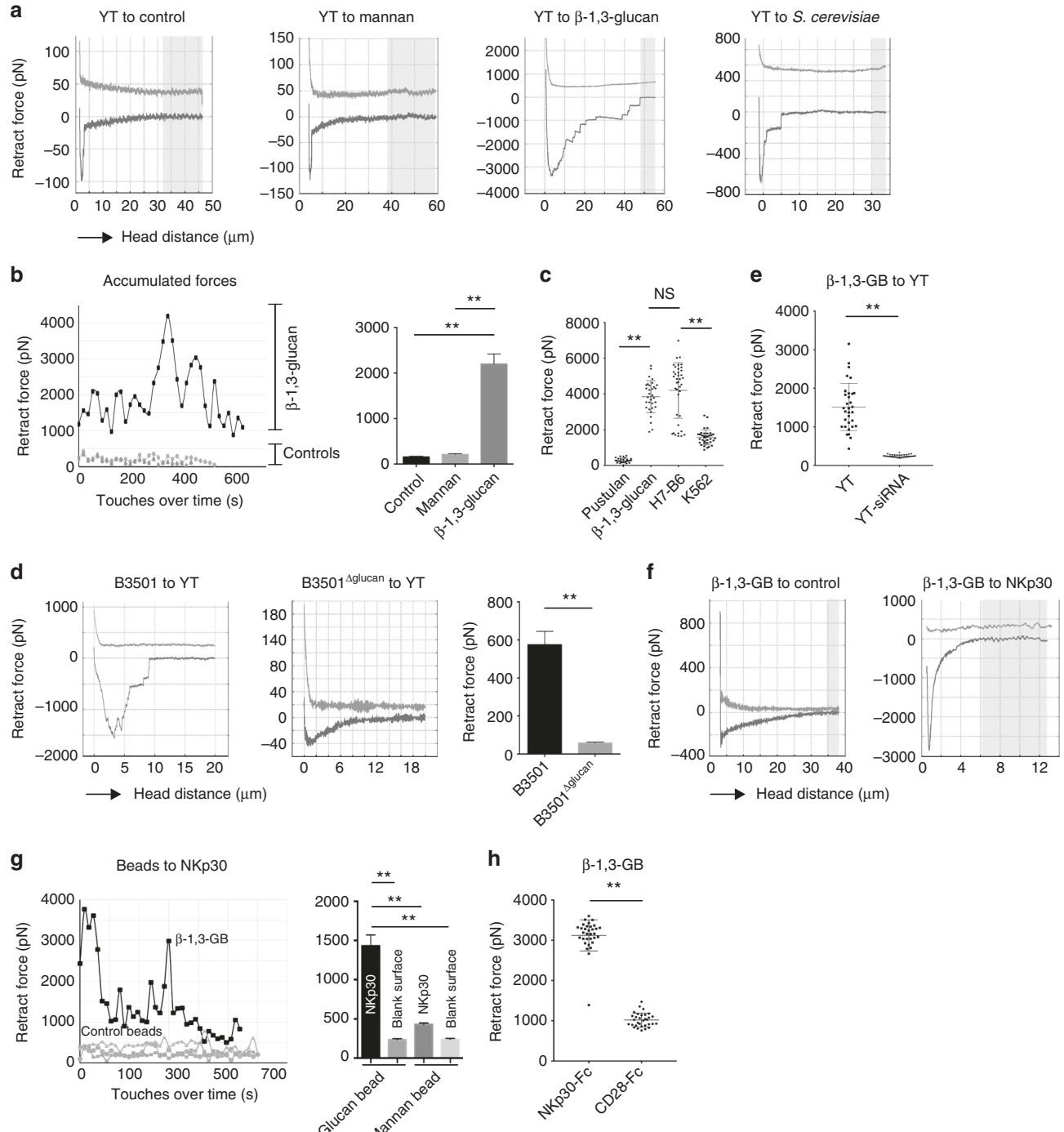

The intensity of NKp30 labeling on TIRF images were up to five times higher in YT cells adhered to β-1,3-glucan than to the control surface (Fig. 5g), suggesting that NKp30 was clustering at the cell surface in response to β-1,3-glucan leading to enhanced NK cell killing.

**β-1,3-glucan activates Src family kinase in NK cells.** Since the Src family kinases (SFK) Fyn and Lyn are required for perforin polarization and NK cell cryptococcal killing[42], we asked whether β-1,3-glucan stimulated SFK activation in NK cells. YT cells were incubated with β-1,3-GB, mannan-conjugated, or unconjugated beads as controls. Using an antibody that detects Tyr416 phosphorylation of SFK, western blot showed activation in YT cells in response to β-1,3-GB, but not to controls (Fig. 6a).

To address whether NKp30 ligation and downstream SFK activation were required for NKp30 clustering, we used 1C01 to block NKp30 binding or used dasatinib to inhibit Src family (Tyr416) activation[43]. Representative photomicrographs of immunocytochemistry (Fig. 6b) and quantitative assessment of the proximity of NKp30 to the synaptic radius (Fig. 6c) showed that NKp30 clustering was abrogated by either 1C01 or dasatinib compared to isotype-matched IgG or DMSO controls (Fig. 6c,d), indicating that NKp30 clustering required NKp30 and SFK signaling.

Live cell imaging was used to determine whether SFK signaling was required for granule polarization in response to β-1,3-glucan. Lysotracker-labeled YT-MAP4-GFP cells were pretreated with 1C01 to block NKp30, or with dasatinib to interrupt SFK signaling. Video images showed that granule polarization at the IS was abolished by treatment with either 1C01 (Fig. 6e, upper panel, Fig. 6f, left panels, Supplementary movie 4) or dasatinib (Fig. 6e, middle panel, Supplementary movie 5) compared to control (Fig. 6f, lower panel and Supplementary movie 6, Fig. 6f, right panel). Thus, NKp30 and SFK signaling was required for NKp30 receptor clustering and granule trafficking and polarization at the IS in response to β-1,3-glucan.

## Discussion

We have made four major observations: 1) β-1,3-glucan bound to recombinant NKp30 and to NKp30 expressed on YT cells; 2) Blocking β-1,3-glucan synthesis in *C. neoformans* inhibited NK cell binding and killing of *Cryptococcus*; 3) β-1,3-glucan activated NK cells to increase expression of NKp30 and perforin, induced NKp30 clustering, granule polarization at the IS, as well as perforin release and enhanced killing of *Cryptococcus* and *C. albicans*; and 4) β-1,3-glucan was able to restore defective NK cell cryptococcal killing from HIV-infected individuals.

We had previously identified NKp30 as an Ig-like transmembrane PRR for fungi[15], and now demonstrate that β-1,3-glucan is the fungal PAMP for NKp30. β-glucans have been observed to protect against bacteria, virus, *Candida*, and parasite in experimental models[44–49], and appears to be beneficial for high-risk surgical patients[50–52]. However, most prior studies were focused on innate immunes cells other than NK cells and it is not clear to what extent activation of NK cells contributed to enhanced host defense in the prior studies.

Our results demonstrate a direct mechanism, whereby complex polymers such as laminarin or β-1,3-glucan from *Euglena* crosslink and activate NKp30, although we have not excluded the possibility that other receptors might bind to β-glucan and play a role. Dectin-1, a C-type lectin receptor, is unlikely to play a role since it is found on macrophages, monocytes, dendritic cells, neutrophils, eosinophils, B cells, and a subpopulation of T cells, but not on NK cells[53,54] or YT cells (Supplementary Fig. 5A). Complement receptor 3 is an integrin (CD11b/CD18, $\alpha_M\beta_2$-integrin) expressed by NK cells and used to recognize β-glucan and mediate cytotoxicity against tumor cells[55]. However, our previous studies failed to identify a role for CR3 in NK cell killing of *Cryptococcus*[56].

These studies have possible implications for translation to patients. We previously showed that IL-12 restored cryptococcal killing by NK cells from HIV-infected patients[36]. However, IL-12 is costly and associated with side effects that limit its therapeutic use. Since β-1,3-glucan is generally not in the blood of patients with cryptococcosis[57,58], we might anticipate that NK cells would not have been pre-activated by β-1,3-glucan, and that NK cells would be responsive to therapy. If β-1,3-glucan could activate NK cell in vivo, it might enhance host defense to fungal infection, including HIV-infected patients who are at risk of cryptococcosis. In addition, although caspofungin is not active against *C. neoformans* at clinically relevant concentrations[59], and would not be used for treatment of cryptococcosis, treatment of fungi with caspofungin compromised the effectiveness of NK-mediated killing of *Cryptococcus*, suggesting that inadvertent treatment of cryptococcosis with an echinocandin would not only lack antimicrobial activity, but would also impair host defense to this pathogen.

In summary, we identified β-1,3-glucan as the PAMP for the cytotoxic PRR, NKp30, and identified the mechanism by which β-1,3-glucan triggered and enhanced NK cell killing of fungi. Furthermore, our data showed that β-1,3-glucan can activate NK cells and restore the defective response in NK cells from HIV-infected patient, which may present therapeutic opportunities.

---

**Fig. 3** NKp30 interaction with β-1,3-glucan revealed by AFM. **a** Representative force–distance (F–D) curves of YT cell binding to β-1,3-glucan. A single YT cell was attached to the cantilever tip. Mannan, media, β-1,3-glucan, or *S. cerevisiae* were coated on poly-L-lysine-treated glass. Peak downward deflection is the force of disruption. **b** Binding forces of YT cells to β-1,3-glucan vs. controls. A single YT cell was brought into contact and was detached from the matrix 20–30 times. At least three spots on the glass were tested per YT cell. Left panel: maximum binding forces from each valid contact over time. Right panel: average binding forces of YT cells to β-1,3-glucan, mannan, and blank glass control. **c** Binding forces of YT cell to β-1,3-glucan vs. H7-B6, and pustulan. A single YT cell was brought into contact and was detached from the matrix multiple times. Three to six areas on the coated glass were tested per YT cell. **d** F–D curves of *C. neoformans* binding to YT cells. A single B3501 cell was fixed to a cantilever tip, and YT cells were adhered to either uncoated or poly-L-lysine glass. Left panel: binding of untreated *C. neoformans* strain B350, middle panel binding of caspofungin-treated, β-1,3-glucan-deficient *C. neoformans* (B3501$^{\Delta\beta-glucan}$), right panel average forces of untreated B3501 and B3501$^{\Delta\beta-glucan}$. **e** Binding forces of β-1,3-glucan-coated beads to wild type vs. siRNA-NKp30-treated YT cells. β-1,3-glucan-coated beads (β-1,3-GB) were glued to a cantilever, and the cells were adhered to a poly-L-lysine-coated ibidi dish. **f** Representative F–D curves of binding forces of polystyrene beads conjugated with or without β-1,3-glucan to recombinant NKp30 (right panel) vs. control glass surface (left panel). **g** Accumulated binding forces of beads conjugated with or without β-1,3-glucan to NKp30 or controls (left panel); average binding forces of beads conjugated with or without β-1,3-glucan to NKp30 or controls (right panel), note change of scale on vertical axis. **h** Binding forces of β-1,3-glucan-coated beads (β-1,3-GB) to recombinant Fc chimera NKp30 vs. recombinant Fc chimera CD28. All experiments were repeated two to four times on different days with similar results. Glucan β-1,3-glucan, glucan beads β-1,3-glucan-conjugated polystyrene beads, isotype control IgG, NKp30 polyclonal anti-NKp30 antibody. **, $p < 0.01$. Data were analyzed using one-way ANOVA **b**, **c**, **g** or *T*-test **d**, **e**, **h**. pN, piconewton

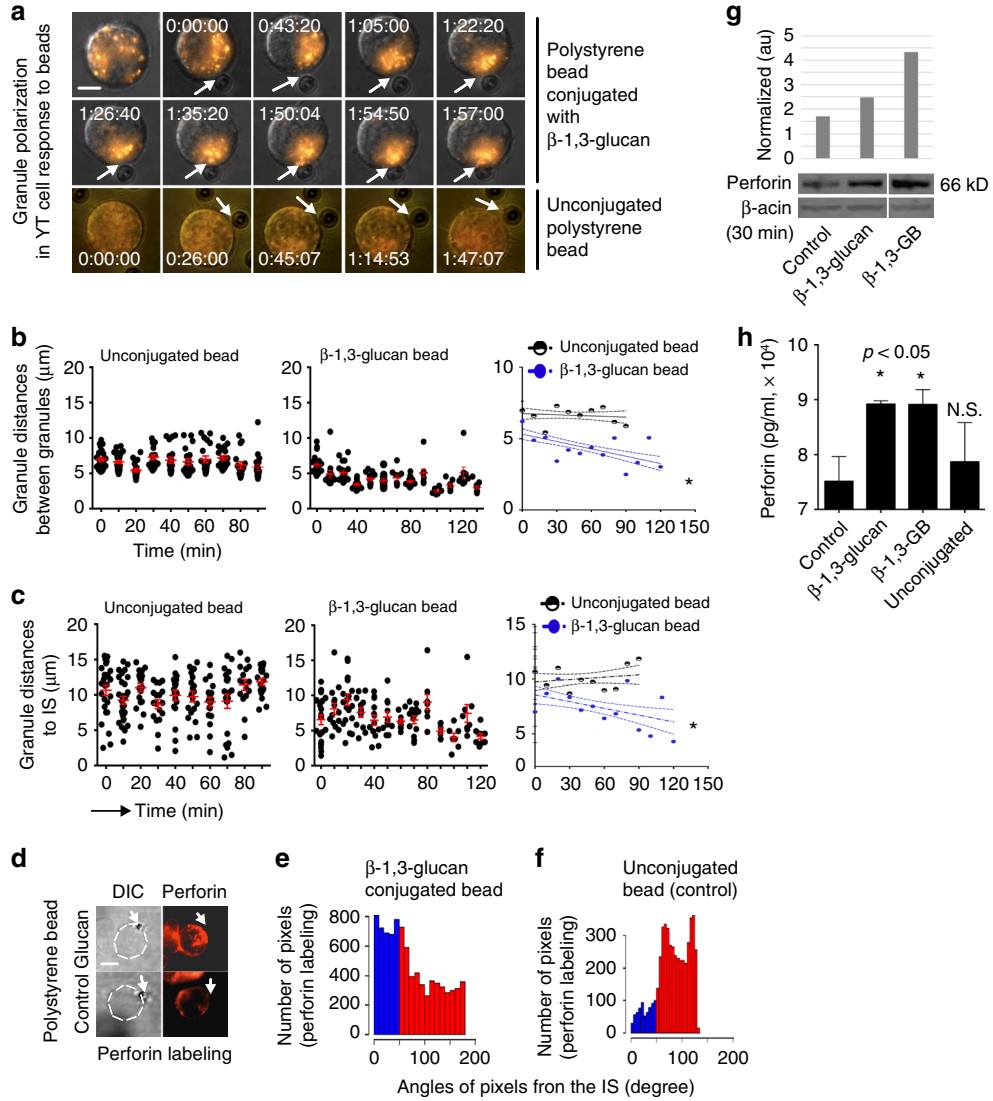

**Fig. 4** β-1,3-glucan stimulates granule polarization in NK cells. **a** Representative images from real-time live cell imaging (Supplementary movie 1) of granule polarization toward the IS in NK cells in response to β-1,3-GB. YT cells were loaded with Lysotracker Red to visualize granules and mixed with β-1,3-glucan-conjugated, or control beads. The experiment was repeated nine times with eight having similar results. Bar = 5 μm. **b** Distance between granules (black circles) and mean distance between granules (red symbols) from panel **a**. The mean distance between granules with 95% confidence intervals is shown in the right panel. **c** Distances between granules and the IS, defined as the center of the point of contact between a YT cell responding to unconjugated beads (left panel) or β-1,3-glucan beads (middle panel) from panel **a**. The mean distance from granules to IS with 95% confidence intervals is shown in the right panel. Distances were determined at times throughout the video and do not necessarily corresponding to times in **a**. **d** Perforin expression and polarization at the IS between YT cells and β-1,3-GB in a representative single YT cell (20 cells analyzed). Experiments were repeated at least three times with similar results. Bar = 5 μm. **e** Proximity of perforin labeling to the radius defined by the center of the IS with the β-1,3-GB. Polarization was defined when the peak of fluorescence was at an angle <45° (blue bars in the left). **f** Proximity of perforin labeling to the synaptic radius with unconjugated bead. Polarization was defined when the peak of fluorescence was at an angle <45° (blue bars in the left). **g** Western blot analysis of perforin expression in YT cells in response to soluble and β-1,3-GB. The experiment was repeated at least three times with similar results. **h** Perforin levels released in the culture media assessed using ELISA. YT cells were treated with soluble β-1,3-glucan, β-1,3-GB, or control. Quadruplicate determinations were performed in two experiments with similar results. N.S. non-significant. *, $p < 0.05$. Data were analyzed using one-way ANOVA (**b**, **c**, **h**)

## Methods

**Cells and microorganisms**. YT cells[60] (NK cell line, a gift from C. Clayberger, Stanford University, Stanford, CA), YT-MAP4-GFP cells were derived from YT cells transfected with microtubule-associated protein 4 (MAP4)-GFP (unpublished data). Primary NK cells were isolated from the blood of healthy and HIV-infected individuals using EasySep (StemCell, #19055) as described[15]. Primary NK cells were from HIV-infected patients receiving antiretroviral therapy with CD4 +counts between 300 and 700 cells/μl and no detectable viral load (< 40 copies/ml) from the Southern Alberta Clinic. K562 cells (ATCC CCL-243) are hematopoietic malignant cells (a gift from Dr. Oliver Bathe, University of Calgary). These cells were maintained in complete RPMI 1640 medium supplemented with 10% FCS, 1% pen-strep, 1% sodium pyruvate (Invitrogen, GIBCO 11360), and 1% non-essential amino acids (all from Invitrogen) at 37 °C and 5% CO$_2$. The recombinant Fc chimera H7B6 was purchased from R&D Systems (8984-B7). Recombinant Fc chimera CD28 was purchased from BioLegend (NM_006139). *C. neoformans* strain B3501 (ATCC 34873) and *C. albicans* (ATCC 58716, serotype A, formerly LUMC 101) were cultured at 32 °C overnight in Sabouraud dextrose broth (BD, Cat. 238230) with shaking until they were at log phase of growth before the experiment. *C. neoformans* strain B3501 was treated with caspofungin (SML0425, Sigma-Aldrich) at 37 °C for 5–6 h to block β-1,3-glucan synthesis (named B3501$^{\Delta glucan}$).

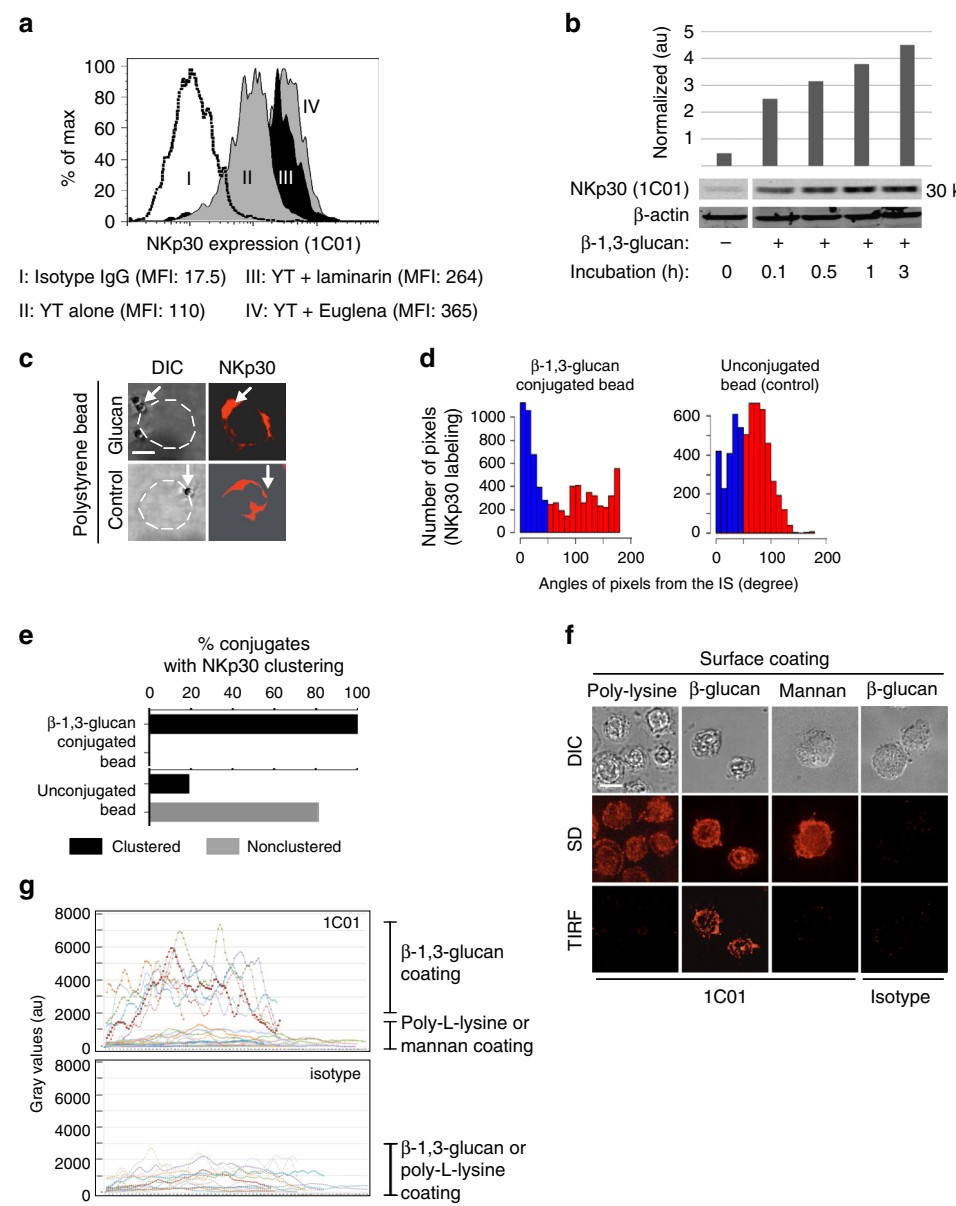

**Fig. 5** β-1,3-glucan increases expression and clustering of NKp30 in NK cells. **a** NKp30 expression in YT cells in response to β-1,3-glucan. YT cells were treated with β-1,3-glucan (III and IV) and expression of NKp30 using 1C01 was compared to untreated cells (I), or untreated cells labeled with isotype control (II). **b** Western blot analysis of NKp30 expression in YT cells in response to β-1,3-glucan. YT cells were incubated with or without β-1,3-glucan for various times. The experiment was repeated three times with similar results. **c** NKp30 labeling at the synapse between a representative single YT cell and a single β-1,3-glucan-conjugated bead. 1C01 was used to label NKp30. The location of the bead is shown with a white arrow. Bar = 5 μm. **d** Proximity of NKp30 expression in relation to the radius defined by the center of the IS ($n = 21$ cells) in panel **c**. The method for determining the angle is illustrated in Supplementary Fig. 6C. Clustering was defined when the peak of fluorescence was at an angle <45° (blue bars in the left). **e** Frequency of NKp30 clustering of all YT cell conjugates with β-1,3-glucan-conjugated or unconjugated beads as analyzed in panel **c**, as defined in panel **d**. **f** TIRF microscopic analysis of NKp30 expression at the interface between YT cells and β-1,3-glucan or control. YT cells were loaded into a glass chamber that had been coated with β-1,3-glucan or mannan incubated at 37 °C for 30 min, fixed and stained for NKp30 using 1C01. Bar = 10 μm. **g** Fluorescent intensity of NKp30 from panel **f** in arbitrary units. Distance is defined in Supplementary Fig. 6A-B. All experiments were repeated three times with similar results. TIRF total internal reflection fluorescence, poly-L-lysine coating alone or mannan coating served as control, Au arbitrary unit, Arrow bead, SD spinning disc microscopy, DIC differential interference contrast digital image, IS immunological synapse

The reduced levels of β-1,3-glucan in B3501$^{\Delta glucan}$ were confirmed using anti-β-1,3-glucan antibody (Biosupplies #400-2, Australia) and flow cytometric analysis. Polystyrene beads conjugated with β-1,3-glucan were prepared as described[33]. Sources for β-1,3-glucan: *S. cerevisiae* (Sigma G5011); barley (Sigma, G6513); *Euglena gracilis* (Sigma, 89862); laminarin (Sigma, L9634); laminarihexaose from curdlan (Megazyme, O-LAM6); and pustulan (Carbomer, 4-00507). Monoclonal antibodies were 1C01 to NKp30[15], anti-NKp30 antibody (R&D, clone# 210845), anti-perforin antibody (eBioscience, Clone eBioBOR21), and anti-β-1,3-glucan

(Biosupplies #400-2, Australia). Rabbit anti-phospho-Src family kinase (Cell Signaling Y416); mouse anti-Fyn (BD Transduction Laboratories).

**Detection of β-1,3-glucan and NKp30 by flow cytometry**. For NK cell surface detection of β-1,3-glucan, YT cells were incubated with a preparation of cryptococcal cell wall/membrane (CCW/M) or purified β-1,3-glucan (e.g., laminarin, laminarihexaose, curdlan) or controls (pustulan) at 4 °C overnight and detected by

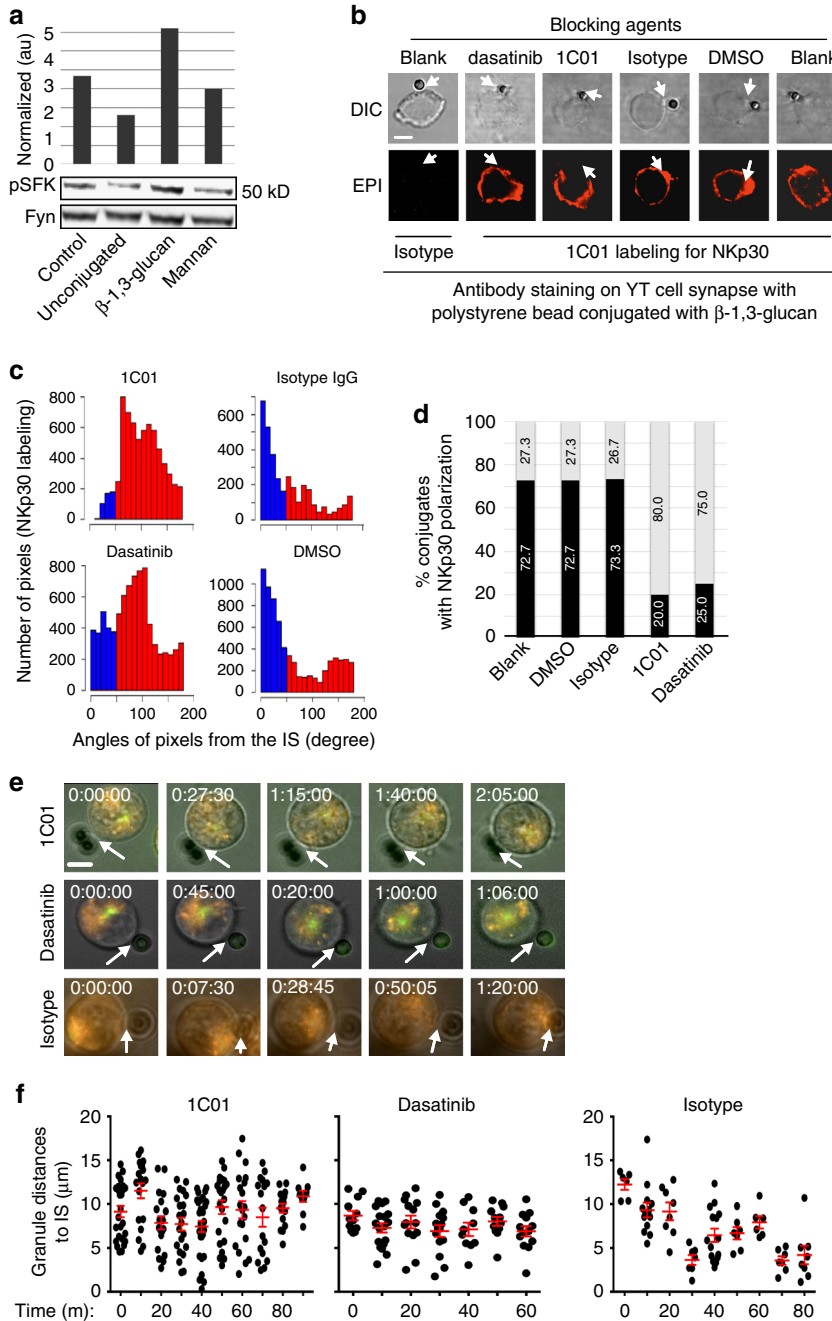

**Fig. 6** β-1,3-glucan induces SFK signaling in NK cell response. **a** Western blot analysis of SFK phosphorylation in YT cells in response to β-1,3-GB vs. controls. The experiment was done twice with similar results. pSFK phosphorylated SFK (Src Family Kinase, Y416), control YT cells alone. **b** NKp30 expression in YT cells in response to β-1,3-GB in the presence of 12.5 µg/ml of 1C01, 50 nM dasatinib, or controls (isotype IgG or DMSO). Cells and beads were labeled with 1C01. Arrow indicates interface between the YT cell and beads. The experiment was repeated at least three times on different days with similar results. Bar = 5 µm. **c** Proximity of NKp30 expression in relation to the radius defined by the IS (representative of n = 15 cells analyzed for 1C01 or 20 cells for dasatinib) in panel **b**. Clustering was defined when the peak of fluorescence was at an angle less than 45° (blue bars in the left). **d** Frequency of NKp30 clustering in YT cell that had formed conjugates with beads in panel **b** as defined in panel **c**. **e** Representative photomicrographs derived from time-lapse videos of YT cells in response to polystyrene beads conjugated with β-1,3-glucan in the presence or absence of 1C01 or dasatnib (Supplementary movies 4–7), respectively. YT cells were loaded with Lysotracker and incubated with 1C01, dasatinib, or control medium before being incubated with β-1,3-GB. The experiment was performed for three times with similar results. Bar = 5 µm. **f** Distances from each granule to the IS derived from data as in panel **a** (red symbols indicate mean ± SEM)

mAb to β-1,3-glucan (Biosupplies, #400-2, Australia). Flow cytometric analysis was performed using Guava easyCyte Flow Cytometer (Guava inCyte v5.2, Millipore Sigma). For flow cytometric analysis of NKp30 binding to β-1,3-glucan, β-glucan-conjugated beads were incubated with a recombinant NKp30-Fc fusion protein (R & D Systems, # O95944) at 4 °C overnight, NKp30-Fc on the beads was detected with anti-NKp30 antibody (1C01). For NKp30 binding to *C. neoformans* and *C. albicans*, the fungi were incubated with a recombinant ectodomain NKp30[61] at 4 °C overnight, and detection of NKp30 was performed using a polyclonal anti-Kp30 antibody (Abnova, PAB17794).

**Immunoprecipitation assay and western blot**. Protein G beads (Dynabeads 100.07D, Thermo Fisher) were used to immunoprecipitate NKp30. Briefly, YT cells were lysed and incubated with β-1,3-glucan (laminarin) at 4 °C for 1–3 days. The protein G beads were conjugated with antibody against β-1,3-glucan or isotype antibody at room temperature for 10 min. These beads were incubated with the mixture of YT cell lysate with β-1,3-glucan at room temperature for 20 min. The immunoprecipitates (complex of NKp30 with its ligand) were eluted from the beads and resolved in NuPAGE Novex 4–12% Bis-Tris Gel (Thermo Fisher, NP0321BOX).

For SFK signaling studies, YT cells were treated with beads conjugated with or without β-1,3-glucan at 37 °C for 5 min, or for the indicated times with soluble β-1,3-glucan. Cell lysates were collected and resolved in NuPAGE Novex 4–12% Bis-Tris Gel (Thermo Fisher). Proteins in the gel were transferred to a nitrocellulose membrane and blotted for NKp30 with 1C01 or anti-phosphorylated SFK (pSFK, Y416).

**Antifungal activity assay**. Unless otherwise specified, $2 \times 10^5$ YT cells or primary NK cells were co-cultured with *C. neoformans* or *C. albicans* at a starting effector to target ratio of 100–200:1 in 200 µl complete RPMI 1640 medium per well in 96-well plates (Costar) at 37 °C. Colony forming units (CFU) were determined as previously described[15]. To investigate the effects of 1C01 or dasatinib (Sigma-Aldrich, CDS023389) on the YT cell response to β-1,3-GB, mannan, or unconjugated beads, YT cells were pretreated with 1C01 or dasatinib at 37 °C for 30 min. Mouse IgG (Life Technologies, 08-6599) or mouse IgG2aκ (BD, 555573) were used as a control for mAb 1C01, rabbit polyclonal IgG (ab27472) was the control for rabbit polyclonal anti-NKp30 (Abnova, PAB17794).

**Single-cell force spectroscopy**. Experiments to determine binding forces using Atomic Force Microscopy (AFM)-based SCFS were performed using a temperature-controlled incubator at 37 °C, supplemented with 5% $CO_2$. A JPK Cellhesion unit (JPK, Berlin, Germany) was used to determine the binding forces. A glass was coated with 0.01% poly-L-lysine and β-1,3-glucan or mannan was loaded into the chambers and air dried. Depending on the experiment, a single YT cell or one to six polystyrene beads were attached to a cantilever tip (320 µm long and 22 µm wide with spring constant of 0.03 N/m) with Cell-Tak (Corning, #354240). Each cantilever with cells or beads was calibrated against the blank control glass dish surface to analyze its thermal fluctuation. All measurements were performed using JPK NanoWizard II with the Cellhesion module (JPK Instruments AG). Briefly, the cantilever was lowered onto a glass dish containing a thin layer of Cell-Tak. The cantilever was moved away from the Cell-Tak while touching the dish surface to wipe off the majority of the Cell-Tak, but retaining enough to glue a YT cell or beads to the tip. The cantilever was then moved to a different dish and gently lowered until the beads or cells were touching the Cell-Tak and fixed on the tip. The recombinant ectodomain of NKp30, β-1,3-glucan, or mannan was coated onto a poly-L-lysine-coated dish. The YT cell or bead(s) on the tip were submerged in 1 ml complete RPMI in the SCFS incubator throughout the experiment. The cantilever tip with the attached cell or bead(s) was lowered toward, touched, and lifted off the β-1,3-glucan or NKp30-coated glass surface (IP gain: 5 Hz; IG gain: 0.0002 Hz; correct baseline: 1; relative set point: 0.16-0.5 nN; z length: 10µm; extend time: 3 s; extend delay: 3–7 s; constant height, extend/retract speed: 10 or 5 µm/s, respectively) 20–30 times. Retraction force between the single YT cell and the β-1,3-glucan surface or fixed bead(s) on the cantilever and NKp30-coated surface on the dishes was recorded. Measurements were repeated on more than three different areas of the coated glass dish. Representative F–D curves were used to show the retraction force vs. distance. Retraction force was the force used to detach the cell or beads on the cantilever tip from the coating on the glass dish, and distance was the distance between the tip and the coating on the dish. F–D curves were normalized and analyzed using JPK Data Processing software (JPK Instruments AG).

**Fluorescence microscopy and live cell imaging**. For fluorescent microscopy, 1C01 was used to label NKp30 for immunocytochemistry and Lysotracker Red DND-99 (ThermoFisher, L7528) was used to label granules for live cell imaging. Briefly, YT cells were incubated with 5 µl mAb 1C01 (2.5 µg/ml) on ice for 30 min and the labeling was visualized using a goat anti-mouse antibody conjugated with Alexa 555 (red, Molecular Probes, A-21422). To block NKp30 and assess granule (Lysotracker positive) movement, YT cells were pretreated with 25 µl 1C01 or an equivalent amount of isotype IgG at 37 °C for 30 min before polystyrene beads (stimulus or control) were added and co-cultured at 37 °C for the entire period of experiment. The cells were mounted onto glass slides, fixed, and labeled with secondary antibody against 1C01, anti-perforin-FITC antibody (BD 556577, green), and DAPI. Conjugate formation was determined using differential interference contrast (DIC) imaging. Both DIC and fluorescent imaging were performed using Delta Vision microscopy (Applied Precision Inc.) with a PlanApo ×60 objective (1.42 NA) equipped with stacking capabilities. DIC and fluorescence images represented one de-convoluted Z-stack obtained using the digital deconvolution program, SoftWoRx v3.5.1, API (Issaquat, WA, USA). Representative images were contrast-enhanced for clarity using SoftWoRx v3.5.1.

**Total internal reflection fluorescence microscopy**. For TIRF imaging, a Leica DMI6000 B inverted microscope was used, which has adaptive focus control, multi-modal optical head, and fast spinning disk (SD) confocal capacity. An 8-chamber dish was pre-coated with 0.01% poly-L-lysine, and β-1,3-glucan or mannan was loaded into the chambers and air dried. YT cells were then loaded into the dish chambers and incubated at 37 °C for 4 h. The cells were fixed with 1% formaldehyde and stained with 1C01, and the labeling was visualized with a goat anti-mouse secondary antibody conjugated with Alexa 555 (Invitrogen). To capture the labeling only at the IS at the cell surface, a depth of 80–100 nm from the lens was set and imaged at high magnification (×63, oil immersion). SD microscopy was used to visualize NKp30 labeling throughout the YT cell body, including the IS. Mannan and poly-L-lysine coating alone served as controls.

**Imaging quantifications**. Velocity image analysis software (Perkin Elmer, Waltham, MA) was used to analyze the movies and quantify the Lysotracker-stained granules trafficking and determine distances to IS. ImageJ (NIH) was used to quantify the NKp30 staining intensities in TIRF microscopy, and a program was written to quantify the distribution of NKp30 and perforin in cells, the location of fluorescent pixels in relation to the IS are described. The point of contact of YT cell with the bead was manually established and set as a reference point to indicate the synapse. The "find spot" measurement function was used to identify granules. Background staining was excluded by adjusting spot intensity to identify only fluorescent granules. For measurements of granule distances, spot intensity was manually offset by 50% with the brightest spot within a radius of 0.5 µm. Distances of the granules to the synapse were determined by measuring the distance from the centroid of each granule in the YT cell to the point of contact with the bead over time.

For ImageJ (NIH) quantification, a quadrilateral was drawn over the edges of each cell encompassing the whole cell body in the images and gray values of each pixel in the quadrilateral were measured by the plotting profile function in ImageJ as illustrated (Supplementary Fig. 6A). The gray values of each cell in the images were measured and a graphic figure (Supplementary Fig. 6B) was created with the vertical axis as gray values in arbitrary units (au). All the values from all the cells were compiled to generate the figures (Fig. 5g).

To quantify the distribution of NKp30 and perforin in cells, the location of fluorescent pixels in relation to the IS were measured by calculating the angles of each pixel in relation to the radius from the center of the cell to the center of the IS as illustrated (Supplementary Fig. 6C). The IS was determined manually as the mid-point of the interface between YT cells and beads. The center of the cell was determined manually. Another line was drawn between pixels with fluorescence above a threshold of 0.2 (NKp30) or 0.4 (perforin) corresponding to 10% of the full dynamic range of the 12-bit camera and the center of the cell. An illustration in Supplementary Fig. 6C showing an angle $\alpha$ <45° or β, greater than 45° between these two lines was determined for each pixel. For the analysis, a program written in R language[62] using RStudio[63] with libraries from EBImage[64] was used to identify each fluorescent pixel in each image. The number of the pixels above a set threshold was plotted against their respective angles from the IS (histograms). NKp30 clustering or perforin polarization was defined when the peak of positive pixels was at an angle of <45° to the IS.

**Statistics**. Statistical studies were performed using GraphPad Prism (GraphPad Software, Inc. La Jolla, USA). One-way ANOVA with Bonferroni comparison tests with Welch correction or unpaired $T$-test (two-tailed) was used to evaluate differences among conditions with error bars of SEM unless otherwise specified. In all cases $p < 0.05$ was considered significant.

**Study approval**. All recruited volunteers provided written informed consent. Use of human materials was approved by The Conjoint Health Research Ethics Board, and all experiments including samples from HIV-infected patients were performed in a BSL2+laboratory.

**Data availability**. The authors declare that the data supporting the findings of this study are available within the article and its supplementary information files, or are available upon reasonable requests to the authors.

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

## Acknowledgements

This work was supported by a grant from the Canadian Institutes for Health Research (CIHR) (CHM #365812), the Jessie Bowden Lloyd Professorship in Immunology (CHM), by an equipment and infrastructure grant from the Canadian Foundation for Innovation (CFI), the Alberta Science and Research Authority, and the Jessie Bowden Lloyd Professorship (S.S.L.). Work with immunofluorescence microscopy and Leica DMI6000 B Inverted research level microscope were supported by the Live Cell Imaging Facility, funded by the Snyder Institute for Chronic Disease, University of Calgary, with invaluable assistance from Dr. Pina Colarusso and Ms. Jennifer (Amon) Poirier. The authors extend thanks to Dr. M. John Gill who provided blood samples from HIV-infected patients and to Danuta Stack and Martina Timm-McCann for their expert technical assistance.

## Author contributions

S.S.L. conceived the concepts, designed the overall study, planned and executed experiments, performed data analyses, and wrote the manuscript; H.O. performed the microscope imaging and analysis; M.K.M. and J.M.V prepared the polystyrene beads conjugated to β-1,3-glucan or mannan. R.X. did the signaling studies, wrote the software, and analyzed staining distribution; L.S. performed part of the fungal killing assay; R.M. created the recombinant ectodomain NKp30 protein; F.M., P.M. and M.A. provided expert support on AFM; S.M.R. provided intellectual advice and review of the manuscript; and C.H.M. provided intellectual input, discussed results, and edited the manuscript. All authors reviewed the manuscript.

## Additional information

**Competing interests:** The authors declare no competing financial interests.

