## [Peer Review File · Nature Communications]

Reviewers' comments:

Reviewer #1 (Atomic force microscopy on immune cells)(Remarks to the Author):

Though NKp30 was a known NK activating receptor, NKp30 was recently identified to be a novel pattern-recognition receptors (PRR) by the authors of this manuscript. However, a microbial pathogen-associated molecular patterns (PAMP) for NKp30 remained to be identified. In this manuscript, β -1,3-glucan was identified as the PAMP for NKp30, and the related mechanisms by which β -1,3-glucan triggered and enhanced NK cell killing of fungi were proposed. Moreover, β -1,3-glucan was demonstrated to activate NK cells and restore the defective response in NK cells from HIV-infected patient.

The data and the analysis are convincing. The results of single cell force spectroscopy are reasonable. The conclusions are sound. It is believe that the results are interesting to relevant readers.

- 1) A question is why soluble β -1,3-glucan can enhance fungal killing and restore defective cryptococcal killing by NK cells from HIV positive individuals?
- 2) What is the spring constant of AFM cantilever?
- 3) In page 10 (the third line from the bottom), Fig. S3F should be corrected to Fig. S2F.
- 4) Primary NK cells were isolated from the blood of HIV-infected individuals. A concern is how to ensure the safety of experimental operators and avoid infection from HIV. What is the level of the laboratory for biosafety protection?

Reviewer #2 (General NK receptor/biology)(Remarks to the Author):

The authors show that the interaction between NKp30 and β -1,3-glucan is involved in the recognition of *Cryptococcus neoformans* and *Candida albicans* by NK cells. While interesting, the authors should address the following points to make data fully convincing.

1. In Fig.1, many controls are missing to confirm the specificity of this interaction.
2. To determine whether the NKp30 receptor is required for this interaction, both NKp30+ and NKp30-YT cells should be tested. This could be easily performed by knocking-down NKp30 expression using shRNA or CRISPR/Cas technologies.
3. It is of interest to show whether NKp30 interacts only with β -1,3-glucan : others β -glucans must be included as controls.
4. The authors need to reinforce data regarding the direct interaction between NKp30 and β -1,3-glucan (Fig.1D,F). First, this interaction must be analyzed not only with NKp30 coated beads but also by another assay such as SPR. In these assays, the specificity of the binding of NKp30-Fc to β -1,3-glucan must be compared to results obtained with at least one irrelevant Fc protein and other β -glucans.
5. In Fig2, it is shown that β -1,3-glucan treatment enhances NK cell killing. The killing is slightly increased but as as mentioned previously there is no formal evidence that this effect is due to the β -1,3-glucan-NKp30 interaction. NKp30-negative YT cells must be used as control cells.
6. In Fig2, it must be demonstrated that high concentration of any glucan (50 and 500 μ g/ml) is not able to increase YT or NK cell activity.
7. The strength of this interaction was quantified using an atomic force microscope in Fig3. It is

comparable to LFA-1 and ICAM-1 interaction. However, it would have been better to compare it with the interaction of NKp30 with another well characterized ligand (such as B7H6).

8. Live cell imaging data need again supplementary controls. First, unconjugated polystyrene beads should be replaced by irrelevant sugar-coated beads. Second, NKp30-negative YT cells must be used as control cells.

Reviewer #3 (Innate response, fungal infection)(Remarks to the Author):

Li et al. previously reported that NKp30 plays an important role for NK cells to recognize and kill *Cryptococcus neoformans* and *Candida albicans* (Li et al. Cell host & Microbe, 2013). In this study, the authors investigated the mechanisms of NKp30 mediated recognition and killing of these fungi. The authors described that β -1,3-glucan bound to NKp30 based on the analysis with flow cytometry and live cell atomic force microscopy. The authors demonstrated that β -1,3-glucan enhanced NK cell killing of *Cryptococcus* through increased release of perforin. Interestingly, β -1,3-glucan increased NKp30 expression and clustering at the microbial and NK cell synapse. The authors concluded that they identified the fungal PAMP that triggers cytotoxic pattern recognition receptor (PRR)-mediated NK cell killing of *C. neoformans* and *C. albicans*.

This study is an interesting and well-presented work. The authors performed a variety of experiments and provided data to indicate their conclusion. However, a couple of additional experiments are needed to confirm their conclusion.

Major Comments:

1.The authors demonstrated that β -1,3-glucan enhanced NK cell killing of *C. neoformans* and *C. albicans* (Figure 2). The authors described that β -1,3-glucan bound to ectodomain of NKp30 based on the analysis with flow cytometry and live cell atomic force microscopy. However, it was not shown whether enhanced NK cell killing of fungi by β -1,3-glucan stimulation was mediated by NKp30. To confirm that NKp30 on NK cells functions as a PRR for β -1,3- glucan, the authors should examine whether knocking down or inhibiting NKp30 with 1C01 would diminish the β -1,3-glucan-mediated enhanced NK cell killing of fungi. Both YT cells and primary NK cells should be used for this experiment.

The title of this manuscript is "Identification and real-time imaging of the fungal PAMP that triggers cytotoxic pattern recognition receptor-mediated NK cell killing of *Cryptococcus neoformans* and *Candida albicans*". However, *C. albicans* was used in only one figure (Figure 2B). To make this title appropriate, the authors should perform the experiments suggested above with both *C. neoformans* and *C. albicans*.

2.The authors showed that the expression and clustering of NKp30 is increased by stimulation with β -1,3-glucan (Figure 5). These features are interesting and this mechanism may be important for activation of NK cells to kill fungi. However, these results raise a possibility that recognition of β -1,3-glucan by other PRRs might had lead to recruitment of NKp30 to the synapse. To confirm that NKp30 is a PRR for β -1,3- glucan, the authors should examine whether knocking down or inhibiting NKp30 with 1C01 would diminish the increased expression and clustering of NKp30 by β -1,3-glucan stimulation.

3.Dectin-1 is also known as a PRR for β -1,3-glucan. Since the authors are trying to show that NKp30

is a new PRR for β -1,3-glucan, it is important to show that dectin-1 is not expressed in YT cells by PCR and staining with anti-dectin-1 Ab even if the previous studies indicated that NK cells do not express dectin-1. If YT cells express dectin-1, the authors should examine whether dectin-1 is involved in increased expression and clustering of NKp30 and enhanced killing of fungi by NK cells stimulated with β -1,3-glucan.

Minor Comments:

1. Figure 4H: There are two " β -1,3-glucan", but one of them should be " β -1,3-GB".

Responses to Reviewers' comments

We thank the 3 referees for indicating that our work is of considerable potential interest and for providing comments, which we believe have substantially strengthened the manuscript. We would also like to thank you for the extension so that we could perform the requested experiments. We provide a point-by-point response to the referees' comments and a completed checklist.

Reviewer #1 (Atomic force microscopy on immune cells)(Remarks to the Author):

Though NKp30 was a known NK activating receptor, NKp30 was recently identified to be a novel pattern-recognition receptors (PRR) by the authors of this manuscript. However, a microbial pathogen-associated molecular patterns (PAMP) for NKp30 remained to be identified. In this manuscript, β -1,3-glucan was identified as the PAMP for NKp30, and the related mechanisms by which β -1,3-glucan triggered and enhanced NK cell killing of fungi were proposed. Moreover, β -1,3-glucan was demonstrated to activate NK cells and restore the defective response in NK cells from HIV-infected patient.

The data and the analysis are convincing. The results of single cell force spectroscopy are reasonable. The conclusions are sound. It is believe that the results are interesting to relevant readers.

Comment: *1). A question is why soluble β -1,3-glucan can enhance fungal killing and restore defective cryptococcal killing by NK cells from HIV positive individuals?*

Response: We have demonstrated that β -1,3-glucan enhanced NKp30 expression and clustering (Figure S5B-C), and perforin release (Figure 4). Previously we showed impaired NKp30 expression and perforin expression and release in NK from HIV-infected patients (Cell Host Microbe. 2013;14:387). We also showed that IL-12 restored NKp30 recognition of and binding to *C. neoformans*, enhanced perforin expression and release, and killing by NK cells from HIV patients. We speculate that the mechanism by which β -1,3-glucan enhances killing of *Cryptococcus* by primary NK cells from HIV positive individuals will be different from the mechanism by which IL-12 enhances killing of *Cryptococcus* by NK cells from HIV positive individuals. This is an active area of investigation in our laboratory.

Comment: *2) What is the spring constant of AFM cantilever?*

Response: We have added this information (0.03N/m) on page 19, 1st paragraph.

Comment: *3) In page 10 (the third line from the bottom), Fig. S3F should be corrected to Fig. S2F.*

Response: We thank the reviewer for noticing this error. We have made the correction, now as Fig. S4G, on Page 11, 2nd paragraph from the bottom.

Comment: *4) Primary NK cells were isolated from the blood of HIV-infected individuals. A concern is how to ensure the safety of experimental operators and avoid infection from HIV. What is the level of the laboratory for biosafety protection?*

Response: In compliance with the biosafety requirements at the University of Calgary, the work using blood from HIV-infected patients was performed in a containment level 2plus physical laboratory that is overseen by Occupational Health and Safety and subject to inspection by the Public Health Agency of Canada.

Reviewer #2 (General NK receptor/biology)(Remarks to the Author):

*The authors show that the interaction between NKp30 and β -1,3-glucan is involved in the recognition of *Cryptococcus neoformans* and *Candida albicans* by NK cells. While interesting, the authors should address the following points to make data fully convincing.*

Response: We thank the reviewer for indicating that they found the data interesting.

Comment: 1. In Fig.1, many controls are missing to confirm the specificity of this interaction.

Response: We have included controls, where appropriate, that include mannan derived from *S. cerevisiae*, which has α -1,6-linked backbone with α -1,2-, α -1,3- linked branches, YT cell alone, naked beads, and isotype-matched antibody to anti- β -1,3-glucan. In response to the reviewer's comments, additional experiments were performed using pustulan (β -1,6-glucan) to confirm the specificity of YT cell binding to β -1,3-glucan (Figures 1D, 3C, E and H).

Comment: 2. To determine whether the NKp30 receptor is required for this interaction, both NKp30+ and NKp30- YT cells should be tested. This could be easily performed by knocking-down NKp30 expression using shRNA or CRISPR/Cas technologies.

Response: In response to the reviewer's comment, we performed experiments to knock down NKp30 expression in YT cells. Live cell imaging or atomic force microscopy, showed that YT cells with NKp30 knocked down had no response to β -1,3-glucan-coated beads (Lysotracker labeled granules, Video S7) and did not binding to β -1,3-glucan-coated dish (Figure 3E). Additional loss of function experiments were performed using a monoclonal antibody, 1C01, which blocked NKp30 clustering and perforin polarization (Figure 6C) in response to β -1,3-glucan. Taken together, our data showed that NKp30 is required for YT cell response to this polysaccharide.

Comment: 3. It is of interest to show whether NKp30 interacts only with β -1,3-glucan: others β -1,3-glucan must be included as controls.

Response: We demonstrated NKp30 did not interact with mannan derived from *S. cerevisiae*, which has α -1,6-linked backbone with α -1,2-, α -1,3- linked branches (Figures 3, 5F, 6A and Video S3). In response to the reviewer's comment, we performed additional experiment using atomic force microscopy to test YT cell binding to pustulan, which is a β -1,6-glucan. These experiments showed no binding to β -1,6-glucan (Figure 3C).

Comment: 4. The authors need to reinforce data regarding the direct interaction between NKp30 and β -1,3-glucan (Fig.1D,F). First, this interaction must be analyzed not only with NKp30 coated beads but also by another assay such as SPR. In these assays, the specificity of the binding of NKp30-Fc to β -1,3-glucan must be compared to results obtained with at least one irrelevant Fc protein and other β -1,3-glucan.

Response: In addition to assessing binding using flow cytometry, we used atomic force microscopy (AFM), which allowed us to assess the interaction under more physiologic temperature and pH. We also assessed the response to a β -1,3-glucan coated surface as well as to β -1,3-glucan coated beads. We believe this has superior to SPR, which would not allow this assessment. We showed strong NKp30 binding to β -1,3-glucan but minimal to control carbohydrates, such as mannan, pustulan and another control Fc fusion protein (CD28-Fc) (Figure 3).

Comment: 5. In Fig2, it is shown that β -1,3-glucan treatment enhances NK cell killing. The

killing is slightly increased but as mentioned previously there is no formal evidence that this effect is due to the β -1,3-glucan-NKp30 interaction. NKp30-negative YT cells must be used as control cells.

Response: To address the reviewer's comment, we performed new experiments in which NKp30 was knocked down using siRNA. Using live cell imaging examining we showed that YT cells with NKp30 knocked down failed to respond to β -1,3-glucan-coated beads (Video S7). Moreover, using AFM, we showed that these YT cells failed to bind to β -1,3-glucan-coated beads (Figure 3E). Furthermore, we previously demonstrated that NKp30-negative YT cells failed to kill fungi due to the lack the receptor NKp30 required for fungal recognition (Cell Host Microbe. 2013 Oct 16;14(4):387-97). These data led us to conclude that YT cell response to β -1,3-glucan or *Cryptococcus* is via NKp30.

In addition, we have demonstrated that 1C01, the monoclonal antibody against NKp30, blocked binding (Figure 1), granule /perforin polarization (Figure 4), and signaling (Figure 6). Previously, we also showed that 1C01 blocked YT cell killing (Cell Host Microbe. 2013 Oct 16;14(4):387-97). These data reveal that the enhanced NK cell killing is due to the interaction between NKp30 and β -1,3-glucan.

Comment: 6. In Fig2, it must be demonstrated that high concentration of any glucan (50 and 500 μ g/ml) is not able to increase YT or NK cell activity.

Response: To address the reviewer's comment, we performed additional experiment using pustulan (β -1,6-glucan) to determine whether this carbohydrate could enhance NK cell killing. Our data showed no enhanced killing by pustulan at similar range of concentrations (Figure S3D). In addition, we have demonstrated that mannan did not interact with YT cells and NKp30, (Figure 3A, B and G, Figure 5F-G), did not stimulate YT cell signaling (Figure 6A), exerted no response using live cell imaging (Video 3).

We acknowledge that we have not exhausted the possibility that other glucans may enhance YT cell killing. However, our data is most consistent with the conclusion that β -1,3-glucan enhanced NK cell killing of *Cryptococcus* and *C. albicans*.

Comment: 7. The strength of this interaction was quantified using an atomic force microscope in Fig3. It is comparable to LFA-1 and ICAM-1 interaction. However, it would have been better to compare it with the interaction of NKp30 with another well-characterized ligand (such as B7H6).

Response: to address the reviewer's comment, we have measured YT cell binding forces to a tumor cell line K562, which express H7-B6, and a recombinant H7-B6 using atomic force microscopy. This data showed comparable binding force between YT cell to β -1,3-glucan and H7-B6 (Figure 3C).

Comment: 8. Live cell imaging data need again supplementary controls. First, unconjugated polystyrene beads should be replaced by irrelevant sugar-coated beads. Second, NKp30-negative YT cells must be used as control cells.

Response: to address the reviewer's comment, we have performed new experiments to knock down NKp30 expression in YT cells and performed live cell imaging and atomic force microscopy. These experiments revealed that YT cells with NKp30 knocked down failed to polarize granules in response to β -1,3-glucan-coated beads (Lysotracker labeled granules, Video

S7), and demonstrated no binding to a β -1,3-glucan-coated surface (Figure 3E). In addition, we have provided data that YT cells had no response to mannan-coated beads (Video S3).

Reviewer #3 (Innate response, fungal infection)(Remarks to the Author):

Li et al. previously reported that NKp30 plays an important role for NK cells to recognize and kill Cryptococcus neoformans and Candida albicans (Li et al. Cell host & Microbe, 2013). In this study, the authors investigated the mechanisms of NKp30 mediated recognition and killing of these fungi. The authors described that β -1,3-glucan bound to NKp30 based on the analysis with flow cytometry and live cell atomic force microscopy. The authors demonstrated that β -1,3-glucan enhanced NK cell killing of Cryptococcus through increased release of perforin. Interestingly, β -1,3-glucan increased NKp30 expression and clustering at the microbial and NK cell synapse. The authors concluded that they identified the fungal PAMP that triggers cytotoxic pattern recognition receptor (PRR)-mediated NK cell killing of C. neoformans and C. albicans.

This study is an interesting and well-presented work. The authors performed a variety of experiments and provided data to indicate their conclusion. However, a couple of additional experiments are needed to confirm their conclusion.

Response: We thank the reviewer for commenting that the work is interesting and well-presented.

Major Comments:

Comment: *1. The authors demonstrated that beta β -1,3-glucan enhanced NK cell killing of C. neoformans and C. albicans (Figure 2). The authors described that β -1,3-glucan bound to ectodomain of NKp30 based on the analysis with flow cytometry and live cell atomic force microscopy. However, it was not shown whether enhanced NK cell killing of fungi by β -1,3-glucan stimulation was mediated by NKp30. To confirm that NKp30 on NK cells functions as a PRR for β -1,3- glucan, the authors should examine whether knocking down or inhibiting NKp30 with IC01 would diminish the β -1,3- glucan-mediated enhanced NK cell killing of fungi. Both YT cells and primary NK cells should be used for this experiment.*

Response: To address the reviewer's comment, NKp30 was knock down in YT cells and live cell imaging and atomic force microscopy was performed. These experiments showed that YT cells with NKp30 knocked down had no response to β -1,3-glucan-coated beads (polarization of LysoTracker labeled granules, Video S7) or to binding to β -1,3-glucan-coated surface (Figure 3E). In addition, we previously showed that knock down NKp30 and blocking NK30 by IC01 abrogated NK cell killing (Cell Host Microbe. 2013 Oct 16;14(4):387-97). Taken together, our data reveals that NK cell killing is through the interaction of NKp30 with β -1,3-glucan.

In the context of this comment, it is important to note that knock down of NKp30 would abrogate recognition and killing of the yeast and therefore it would not be possible to determine whether NKp30 knockdown abrogated pre-activation of YT cells by β -1,3- glucan. YT cells won't be able to kill since they lack the receptor required for the initial recognition of β -1,3- glucan.

Comment: *The title of this manuscript is "Identification and real-time imaging of the fungal PAMP that triggers cytotoxic pattern recognition receptor-mediated NK cell killing of Cryptococcus neoformans and Candida albicans". However, C. albicans was used in only one figure (Figure 2B). To make this title appropriate, the authors should perform the experiments suggested above with both C. neoformans and C. albicans.*

Response: We thank the reviewer for this comment. We performed additional experiments to test the binding of NKp30 to *C. albicans*. Both *C. neoformans* and *C. albicans* bound recombinant NKp30 (Figure 1G). Unfortunately, caspofungin cannot be used as a loss of function approach for *C. albicans* as it either inhibits the growth, or at sub-inhibitory concentrations, unmask, rather than reduces expression of beta glucan (PLoS Pathog. 2006 Apr;2(4):e35. Epub 2006 Apr 28).

Comment: 2. The authors showed that the expression and clustering of NKp30 is increased by stimulation with β -1,3-glucan (Figure 5). These features are interesting and this mechanism may be important for activation of NK cells to kill fungi. However, these results raise a possibility that recognition of β -1,3-glucan by other PRRs might had lead to recruitment of NKp30 to the synapse. To confirm that NKp30 is a PRR for β -1,3- glucan, the authors should examine whether knocking down or inhibiting NKp30 with 1C01 would diminish the increased expression and clustering of NKp30 by β -1,3-glucan stimulation.

Response: We agree with the reviewer. Indeed, we show that treatment with 1C01 blocks receptor clustering (Figure 6C). In addition, we performed new experiments, where NKp30 was knocked down using siRNA. Using live cell imaging we showed that YT cells with NKp30 being disrupted failed to respond, no granule polarization, to β -1,3-glucan-coated beads (Video S7).

Comment: 3. Dectin-1 is also known as a PRR for β -1,3-glucan. Since the authors are trying to show that NKp30 is a new PRR for β -1,3-glucan, it is important to show that dectin-1 is not expressed in YT cells by PCR and staining with anti-dectin-1 Ab even if the previous studies indicated that NK cells do not express dectin-1. If YT cells express dectin-1, the authors should examine whether dectin-1 is involved in increased expression and clustering of NKp30 and enhanced killing of fungi by NK cells stimulated with β -1,3-glucan.

Response: To address the reviewer's comment, we have performed new experiments to determine whether NK cells express dectin-1. The results demonstrated that YT cells do not express dectin-1 (Figure S4E).

Minor Comments:

Comment: 1. Figure 4H: There are two " β -1,3-glucan", but one of them should be " β -1,3-GB".

Response: Thank you. We have changed the labeling on the third bar to read β -1,3-GB.

REVIEWERS' COMMENTS:

Reviewer #1 (Remarks to the Author):

The authors have revised the manuscript according to the comments.

Reviewer #2 (Remarks to the Author):

The manuscript has been greatly improved. I have no additional requests.

Reviewer #3 (Remarks to the Author):

My concerns were addressed in the revised version.